Prognostic values of the core components of the mammalian circadian clock in prostate cancer

Yue Wenchang 1
Du Xiao 2
Wang Xuhong 3
Gui Niu 4
Zhang Weijie 1
Sun Jiale 1
You Jiawei 1
He Dong 1
Geng Xinyu 1
Huang Yuhua 1
Hou Jianquan houjianquan@suda.edu.cn 1
1 Department of Urology, The First Affiliated Hospital of Soochow University , Suzhou , China
2 Department of Radiation Oncology, The First Affiliated Hospital of Soochow University , Suzhou , China
3 Department of Urology, Tongcheng people’s Hospital , Tongcheng , China
4 General Surgery Ward 2, Fengtaixian Hospital of Chinese Medicine , Huainan , China
Zhang Jian
Electronic publication date: 2021 Dec 9
Publication date: 2021
Volume: 9
Electronic Location ID: e12539
Received 2021 Jul 7; Accepted 2021 Nov 4
Copyright: ©2021 Yue et al.
Copyright year: 2021
Copyright holder: Yue et al.
License: This is an open access article distributed under the terms of the Creative Commons Attribution License, which permits unrestricted use, distribution, reproduction and adaptation in any medium and for any purpose provided that it is properly attributed. For attribution, the original author(s), title, publication source (PeerJ) and either DOI or URL of the article must be cited.
License URL: https://creativecommons.org/licenses/by/4.0/

Keywords: Prostate cancer, The core components of the mammalian circadian clock (CCMCCs), Prognosis, Survival, Risk score model

Funding: The authors received no funding for this work.

==============================
Background

Prostate cancer (PC) is one of the most common malignancies in males. Extensive and complex connections between circadian rhythm and cancer were found. Nonetheless, in PC, the potential role of the core components of the mammalian circadian clock (CCMCCs) in prognosis prediction has not been fully clarified.

Methods

We firstly collected 605 patients with PC from The Cancer Genome Atlas (TCGA) and the Gene Expression Omnibus (GEO) databases. Survival analysis was carried out for each CCMCC. Then, we investigated the prognostic ability of CCMCCs by Cox regression analysis. Independent prognostic signatures were extracted for the establishment of the circadian clock-based risk score model. We explored the predictive performance of the risk score model in the TCGA training cohort and the independent GEO dataset. Finally, the relationships between risk score and clinicopathological parameters, biological processes, and signaling pathways were evaluated.

Results

The expression levels of CCMCCs were widely correlated with age, tumor status, lymph node status, disease-free survival (DFS), progression-free survival (PFS), and overall survival (OS). Nine circadian clock genes, including CSNK1D, BTRC, CLOCK, CSNK1E, FBXL3, PRKAA2, DBP, NR1D2, and RORB, were identified as vital prognostic factors in PC and were used to construct the circadian clock-based risk score model. For DFS, the area under the 3-year or 5-year receiver operating characteristic curves ranged from 0.728 to 0.821, suggesting better predictive performance. When compared with T3-4N1 stage, PC patients at T2N0 stage might be benefited more from the circadian clock-based risk score model. Furthermore, a high circadian clock-based risk score indicated shorter DFS (p < 0.0001), early progression (p < 0.0001), and higher 5-year death rate (p = 0.007) in PC. The risk score was related to tumor status (p < 0.001), lymph node status (p < 0.001), and ribosome-related biogenesis and pathways.

Conclusions

The vital roles of circadian clock genes in clinical outcomes were fully depicted. The circadian clock-based risk score model could reflect and predict the prognosis of patients with PC.

Introduction

Prostate cancer (PC) is the most common malignant tumor in the male urinary system (Siegel et al., 2021). An increased incidence rate of PC was found around the world (Siegel et al., 2021; Chen et al., 2015; Jemal et al., 2016). The incidence and mortality rates of PC reached up to 10 per 100,000 and four per 100,000, respectively (Siegel et al., 2021; Chen et al., 2015; Jemal et al., 2016). Individual differences of PC are obvious. For PC, the therapeutic scheme of each patient mainly depends on tumor grade, doctors’ clinical judgment, and conventional risk assessment. Conventional risk assessment included several clinical factors: prostate specific antigen (PSA) level, TNM staging, and Gleason score. Prostatectomy and radiotherapy are the recommended treatments for localized PC. Even the prognosis of most clinical patients with early-stage PC was satisfactory, postoperative recurrence was unavoidable (Seikkula et al., 2018). For advanced PC patients, despite initial sensitivity to androgen deprivation therapy, the majority of patients with PC finally developed resistance to castration therapy after 18 to 24 months of clinical treatment (Seikkula et al., 2018; Small et al., 2019; Saad et al., 2021). Thus, it is of significance to identify patients with high relapse risk.

The discoveries in the circadian rhythm were considered as dramatic breakthroughs in the field of medicine. In 2017, three scientists won the Nobel Prize for their work on the circadian rhythm (Callaway & Ledford, 2017; Burki, 2017). Several studies highlighted the essential role of circadian disruption in multiple biomolecular processes of cancer, including PC and other solid tumors (Sigurdardottir et al., 2012; Wendeu-Foyet & Menegaux, 2017; Viswanathan, Hankinson & Schernhammer, 2007; Stevens et al., 2014; Papagiannakopoulos et al., 2016; Kettner et al., 2016; Innominato et al., 2009; Huisman et al., 2015). The core components of the mammalian circadian clock (CCMCCs) were defined as a group of genes that could regulate human circadian rhythm through regulating RNA expression levels and biological pathways (Takahashi, 2017; Partch, Green & Takahashi, 2014). CCMCCs are composed of a total of 22 genes. These 22 CCMCCs included seven core clock genes (CLOCK, ARNTL, PER1, PER2, PER3, CRY1) and 15 other circadian clock-related genes (CRY2, BTRC, CSNK1D, CSNK1E, CUL1, DBP, FBXL21, FBXL3, NFIL3, NR1D1, NR1D2, PRKAA1, PRKAA2, RORA, RORB, SKP1). CCMCCs predominately promoted many biochemical activities to work in rule and order, thereby maintaining homeostasis. These key circadian clock genes also affected tumorigenesis, tumor growth, metastasis, and clinical outcomes of cancer patients. In tumor-bearing mice, the expression levels of five CCMCCs, including NR1D1, PER1, PER2, ARNTL, and DBP, were downregulated in hepatic metastasis from colorectal cancer when compared with healthy tissue (Huisman et al., 2015). In lung cancer, both PER2 and ARNTL were tumor suppressor genes (Papagiannakopoulos et al., 2016). In PC, overexpression of the clock gene PER1 promoted tumor cell apoptosis (Cao et al., 2009). The complex physically interaction between PER1 and the androgen receptor was also found (Cao et al., 2009). However, there are very limited researches that investigated the vital functions of key circadian clock genes in the pathogenesis and prognosis of PC. In this study, we systematically explored the association between circadian clock and prognosis in PC. Then, we proposed the circadian clock-based risk score and constructed a circadian clock-related prognostic model. The performance of the risk score model was verified in The Cancer Genome Atlas (TCGA) dataset and the independent Gene Expression Omnibus (GEO) dataset. We found correlations between the circadian clock gene signature and several biological functions, signaling pathways, and clinicopathologic features.

Material and Methods

Dataset acquisition from the TCGA and GEO database

We identified suitable public datasets of PC patients in the TCGA (https://tcga-data.nci.nih.gov/tcga/) and GEO (https://www.ncbi.nlm.nih.gov/geo/) Databases. We eliminated datasets without intact gene expression data and prognostic information. Both RNA sequencing (RNA-seq) data and complete clinical annotation for each PC patient were downloaded online. In total, GSE70770 with 112 PC patients (Ross-Adams et al., 2015) and TCGA-PC dataset with 493 PC patients (Liu et al., 2018) were eventually gathered in this study for further analysis. Characteristics of the TCGA cohort and the GEO cohort were summarized in Table S1. All participants gave their informed consent for publication.

Construction of prognostic signature based on clock genes

A total of 22 CCMCCs were obtained from previously published reviews (Takahashi, 2017; Partch, Green & Takahashi, 2014). We applied the COX regression analysis to assess the effects of CCMCCs on clinical prognosis. We selected potential prognosis-related clock genes to construct the circadian clock-based risk score and the prognosis prediction model. The definition of the circadian clock-based risk score was as follows:

The circadian clock-based risk score = Σ λi

where i represents the expression of prognosis-related clock genes, λ is the coefficient that extracted from the COX regression analysis. The final formula of the circadian clock-based risk score was as follows: the circadian clock-based risk score = (0.8*expression value of CSNK1D) + (−1.824*expression value of BTRC) + (−1.7645*expression value of CLOCK) + (0.4555*expression value of CSNK1E) − (1.239*expression value of FBXL3) + (−1.56*expression value of PRKAA2) + (1.325*expression value of DBP) + (0.433*expression value of NR1D2) + (1.049*expression value of RORB). The training subset and the internal validation subset was from the TCGA cohort, while GSE70770 was used as an external validation dataset. The receiver operating characteristic (ROC) analysis was applied to validate the performance of the proposed model. By R package termed “survivalROC”, areas under the ROC curve (AUC) were calculated.

Functional enrichment analysis

Differentially expressed genes (DEGs) between the high and low circadian clock-based risk score groups were recognized by the “limma” R package. DEGs with absolute value change of expression more than 2 and p value less than 0.05 were selected for signal pathway analysis. The Gene Ontology (GO, http://www.geneontology.org/) and the Kyoto Encyclopedia of Genes and Genomes (KEGG, https://www.kegg.jp/kegg/) enrichment analyses were utilized to reveal unique biological processes and signal pathways between the high and low circadian clock-based risk score groups.

Statistical analysis

The survival analysis of each CCMCC and the circadian clock-based risk score was conducted by the Kaplan–Meier (KM) method and the log-rank test. The optimal cutoff point was evaluated by the R package “survminer”. The differences between the two groups were compared with the t-test. All statistical analyses were two-tailed. And p value less than 0.05 was considered as statistical significance. The visualization of all statistical results was performed by the SPSS 22.0 software and the R 4.0.1 software.

Results

Expression profile of CCMCCs in PC

In the TCGA dataset, the gene expression of 22 CCMCCs and clinical features of 493 PC patients from TCGA database were summarized in Fig. 1A and Table S1. Most subjects were under the age of 65 (327/493, 66.3%). In addition, a majority of patients in the TCGA cohort were at N0 stage (342/493, 69.4%). Pathology T3-4 patients also accounted for a large proportion (300/493, 60.9%). The number of patients who received neoadjuvant therapy was also summarized. We further explored the association between CCMCCs’ expression levels and clinicopathological characteristics. Importantly, the expression level of SKP1 was downregulated in PC patients over 65 years old (Fig. 1B). For lymph node (N) stage, CRY1 (p < 0.006), CSNK1D (p < 0.0014), EBXL21 (p < 0.001), PER1 (p < 0.001), PER2 (p < 0.012), PRKAA2 (p < 0.011), and SKP1 (p < 0.007) were differentially expressed between the N0 and the N1 groups (Fig. 1C). Moreover, for tumor (T) stage, ten CCMCCs differentially expressed between the T2 group and the T3-4 group (all p < 0.05, Fig. 1D). In addition, relative high expression levels of BTRC, CRY2, FBXL21, PER1, PER2, and PRKAA2, were found in the T2N0 group when compared with the T3-4N1 group (both p < 0.05, Fig. S1). In the GEO cohort, we further conducted correlation analysis on Gleason grade, PSA level, and expression levels of 22 CCMCCs (Fig. S2). However, regardless of statistical correlation between Gleason grade and NR1D2 (p = 0.011, r = 0.241), CLOCK (p = 0.045, r = −0.192), as well as SKP1 (p = 0.320, r = −0.205), the correlation was weak (Fig. S2). No statistic relationship was found between PSA level, and expression levels of 22 CCMCCs (Fig. S2).

Figure 1 Expression levels of the core components of the mammalian circadian clock (CCMCCs) in prostate cancer and its correlation with clinical factors.

(A) The expression profile of CCMCCs in prostate cancer. (B) The expression level of SKP1 was significantly associated with age (p = 0.0079). (C) Seven CCMCCs differentially expressed between the N0 group and the N1 group (p < 0.05). (D) Ten CCMCCs differentially expressed between the T2 group and the T3-4 group (p < 0.05). Abbreviation: N, lymph node; T, tumor.

Relationship between CCMCCs and prognosis

To evaluate the association between CCMCCs expression and clinical outcomes, we performed the survival analysis. In the TCGA cohort, the sample size of prostate cancer patients who owned disease-free survival (DFS), progression-free survival (PFS), and overall survival (OS) data was 333, 493, and 493, respectively (Figs. 2–6). We found that high expression levels of ARNTL (p = 0.037), CLOCK (p = 0.006), PER2 (p = 0.0051), PER3 (p = 0.013), BTRC (p = 0.026), CUL1 (p = 0.0054), FBXL21 (p = 0.00035), FBXL3 (p = 0.012), PRKAA1 (p = 0.013), PRKAA2 (p = 0.015), and RORB (p = 0.0067) were related to longer DFS (Figs. 2 and 3). On the contrary, overexpression of CRY1 (p = 0.0015), CSNK1D (p = 0.00032), CSNK1E (p = 0.0019), DBP (p = 0.00075), NR1D1 (p = 0.04), and SKP1 (p = 0.017) were associated with poor DFS (Figs. 2 and 3). For PFS, KM survival analysis found that PFS was statistically related to the expression levels of 18 CCMCCs (Figs. 4 and 5). In addition, there were high connections between the expression levels of CSNK1D (p = 0.016), FBXL21 (p = 0.039), NFIL3 (p = 0.037), PER1 (p = 0.035), RORB (p = 0.037) and OS in KM curves (Fig. 6).

Figure 2 The expression levels of five core clock genes were related to disease-free survival (DFS).

(A) ARNTL (p = 0.037); (B) CLOCK (p = 0.006); (C) CRY1 (p = 0.0015); (D) PER2 (p = 0.0051); (E) PER3 (p = 0.013).

Figure 3 The expression levels of 12 circadian clock-related genes were related to disease-free survival (DFS).

(A) BTRC (p = 0.026); (B) CSNK1D (p = 0.00032); (C) CSNK1E (p = 0.0019); (D) CUL1 (p = 0.0054); (E) DBP (p = 0.00075); (F) FBXL21 (p = 0.00035); (G) FBXL3 (p = 0.012); (H) NR1D1 (p = 0.04); (I) PRKAA1 (p = 0.013); (J) PRKAA2 (p = 0.015); (K) RORB (p = 0.0067); (L) SKP1 (p = 0.017).

Figure 4 The expression levels of four core clock genes were related to progression-free survival (PFS).

(A) CLOCK (p = 0.0085); (B) CRY1 (p = 0.0028); (C) PER2 (p = 0.032); (D) PER3 (p = 0.045).

Figure 5 The expression levels of 14 circadian clock-related genes were related to progression-free survival (PFS).

(A) BTRC (p = 0.031); (B) CSNK1D (p = 0.017); (C) CSNK1E (p = 0.00026); (D) CUL1 (p = 0.025); (E) DBP (p = 0.00092); (F) FBXL21 (p = 0.0014); (G) FBXL3 (p = 0.0066); (H) NFIL3 (p = 0.035); (I) NR1D1 (p = 0.027); (J) NR1D2 (p = 0.044); (K) PRKAA1 (p = 0.023); (L) PRKAA2 (p = 0.0051); (M) RORB (p < 0.0001); (N) SKP1 (p = 0.014).

Figure 6 The expression levels of five the core components of the mammalian circadian clock were related to overall survival (OS).

(A) CSNK1D (p = 0.016); (B) FBXL21 (p = 0.039); (C) NFIL3 (p = 0.037); (D) PER1 (p = 0.035); (E) RORB (p = 0.037).

Considering the influence of TNM stage on prognosis, we applied survival subgroup analysis to the T2N0 cohort and the T3-4N1 cohort (Tables S2–S7). Importantly, at both T2N0 stage and T3-4N1 stage, high BTRC, CLOCK, CRY1, FBXL3, PER3, and RORA expression indicated longer DFS (both p < 0.05, Tables S2–S3). It was also worth to mention that high DBP linked to shorter PFS in T2N0 prostate cancer (p = 0.013, Table S4), while the contrary result was found in T3-4N1 stage patients (p = 0.0016, Table S5). For OS, high NR1D1 expression was only significantly related to better prognosis in T2N0 stage patients (p = 0.041, Table S6), while the negative result was found in T3-4N1 stage patients (p = 0.13, Table S7).

Identification of potential prognostic CCMCCs

In the univariate Cox regression analysis, BTRC (p = 0.020136), CLOCK (p = 0.037606), CSNK1D (p = 0.002343), CSNK1E (p = 0.006002), FBXL3 (p = 0.01807), and PRKAA2 (p = 0.028131) were significantly associated with DFS (Table 1). Additionally, among 22 CCMCCs, six genes, including CLOCK (p = 0.042614), CSNK1E (p = 0.029721), DBP (p = 0.002931), NR1D2 (p = 0.041884), PRKAA2 (p = 0.018837), and RORB (p = 0.00098), showed significant relationship with PFS (Table 2). For OS, CSNK1D was found to be potential prognostic factor (p = 0.000651, Table 3). After combining above findings and removing repetitive gene, 9 CCMCCs were left, including CSNK1D, BTRC, CLOCK, CSNK1E, FBXL3, PRKAA2, DBP, NR1D2, and RORB. We proposed a hypothesis that 9 CCMCCs were key prognostic genes in PC and incorporating them could effectively predict the prognosis of PC patients.

Table 1 Cox regression analysis for disease-free survival.

Variables	HR	Lower limit of 95% CI	Upper limit of 95% CI	p	
ARNTL	0.996443	0.989075	1.003865	0.34664	
BTRC	0.998178	0.996644	0.999715	0.020136	
CLOCK	0.997669	0.995476	0.999866	0.037606	
CRY1	1.001943	0.999512	1.004381	0.117348	
CRY2	1.000162	0.999264	1.001061	0.723401	
CSNK1D	1.000571	1.000203	1.000939	0.002343	
CSNK1E	1.00061	1.000175	1.001045	0.006002	
CUL1	1.00047	0.998849	1.002094	0.569861	
DBP	1.001227	0.999326	1.003132	0.205956	
FBXL21	0.991817	0.942722	1.043469	0.751078	
FBXL3	0.998762	0.997736	0.999788	0.01807	
NFIL3	1.000172	0.999762	1.000582	0.411733	
NR1D1	1.00041	0.998809	1.002014	0.615872	
NR1D2	0.999767	0.998967	1.000568	0.56873	
PER1	0.999872	0.99963	1.000115	0.301755	
PER2	0.999433	0.99829	1.000576	0.330692	
PER3	1.000045	0.999185	1.000906	0.917624	
PRKAA1	0.999719	0.999144	1.000295	0.338853	
PRKAA2	0.997956	0.996135	0.999781	0.028131	
RORA	0.998475	0.995557	1.001403	0.306999	
RORB	1.001115	0.999436	1.002796	0.193172	
SKP1	1.000189	0.999839	1.000538	0.289565	
Notes.

Abbreviation HR hazard ratio

P P value for whole

95% CI 95% confidence interval

Statistically significant data were marked with bold and underline.

Table 2 Cox regression analysis for progression-free survival.

Variables	HR	Lower limit of 95% CI	Upper limit of 95% CI	p	
ARNTL	0.99733	0.993508	1.001166	0.172199	
BTRC	0.999311	0.998442	1.00018	0.120076	
CLOCK	0.998806	0.997653	0.99996	0.042614	
CRY1	1.001142	0.999805	1.00248	0.094108	
CRY2	0.999888	0.999376	1.0004	0.667062	
CSNK1D	1.000072	0.999845	1.000298	0.534479	
CSNK1E	1.000302	1.00003	1.000575	0.029721	
CUL1	1.000777	0.99995	1.001604	0.065408	
DBP	1.001326	1.000452	1.002201	0.002931	
FBXL21	0.968937	0.929841	1.009677	0.133184	
FBXL3	0.999498	0.998974	1.000022	0.060412	
NFIL3	1.000149	0.999917	1.000381	0.20746	
NR1D1	1.000643	0.999837	1.001451	0.118107	
NR1D2	1.000433	1.000016	1.00085	0.041884	
PER1	0.999981	0.999866	1.000096	0.747887	
PER2	1.000206	0.999777	1.000635	0.346838	
PER3	1.000287	0.999887	1.000687	0.159504	
PRKAA1	0.999804	0.999484	1.000124	0.230166	
PRKAA2	0.998926	0.998031	0.999822	0.018837	
RORA	1.000184	0.998629	1.001742	0.816508	
RORB	1.00105	1.000425	1.001674	0.00098	
SKP1	1.000025	0.999834	1.000217	0.79528	
Notes.

Abbreviation HR hazard ratio

P P value for whole

95% CI 95% confidence interval

Statistically significant data were marked with bold and underline.

Table 3 Cox regression analysis for overall survival.

Variables	HR	Lower limit of 95% CI	Upper limit of 95% CI	p	
ARNTL	1.002023	0.9902	1.013987	0.738608	
BTRC	0.999165	0.996398	1.00194	0.554846	
CLOCK	0.997808	0.993549	1.002084	0.314471	
CRY1	0.999584	0.995035	1.004154	0.858089	
CRY2	1.000068	0.998619	1.001518	0.926872	
CSNK1D	1.001029	1.000437	1.001622	0.000651	
CSNK1E	0.999911	0.998903	1.000919	0.862001	
CUL1	1.000169	0.997174	1.003174	0.912052	
DBP	0.997995	0.993311	1.002702	0.403177	
FBXL21	0.853599	0.623965	1.167744	0.322149	
FBXL3	1.000268	0.99868	1.001858	0.741299	
NFIL3	0.999673	0.998706	1.000642	0.508415	
NR1D1	0.997856	0.993634	1.002095	0.321049	
NR1D2	0.999837	0.998527	1.001149	0.807904	
PER1	0.999599	0.998984	1.000215	0.201657	
PER2	0.999363	0.997433	1.001297	0.518096	
PER3	0.999276	0.997557	1.000999	0.410001	
PRKAA1	0.999791	0.998838	1.000744	0.666887	
PRKAA2	1.000341	0.998365	1.002321	0.735204	
RORA	0.997031	0.991089	1.003009	0.329597	
RORB	0.995276	0.987314	1.003302	0.247869	
SKP1	0.99986	0.999268	1.000452	0.642982	
Notes.

Abbreviation HR hazard ratio

P P value for whole

95% CI 95% confidence interval

Statistically significant data were marked with bold and underline.

Construction and validation of circadian clock-based risk score

To verify our hypothesis, we separated the TCGA cohort into the test set and the internal validation set in a ratio of 7:3. All enrolled patients in the cohort underwent surgical treatment, thus we mainly investigated the application of the predictive model in DFS prediction. As shown in Figs. 7A–7B, the AUC value of the predictive model in the training cohort and the validation cohort for 3-year DFS was 0.742 and 0.821, respectively. The ROC curves and AUC values (0.728 for the test cohort; 0.753 for the validation cohort) further indicated the satisfactory predictive power of the circadian clock-based signature in DFS prediction (Figs. 7C–7D). PC patients with high circadian clock-based risk score also had lower DFS than the low risk score group (p < 0.0001, Fig. 7E). In an independent cohort with 112 PC patients who prostatectomy, we divided the GSE70770 cohort into high and low risk score groups and conducted the KM survival analysis. Results showed that the high circadian clock-based risk score was correlated with early relapse (vs low circadian clock-based risk score: 13.000 months vs 21.000 months, p = 0.06; Fig. S3).

Figure 7 Validation of proposed circadian clock-based risk score model in disease-free survival (DFS) prediction by receiver-operator characteristic (ROC) analyses.

(A–B) ROC curves in the training cohort (AUC = 0.742) and the validation cohort (AUC = 0.821) for 3-year. (C–D) ROC curves in the training cohort (AUC = 0.728) and the validation cohort (AUC = 0.753) for 5-year. (E) High circadian clock-based risk score was correlated with shorter DFS (p < 0.0001).

Moreover, we explored the impact of TNM stage. When compared with clinical prognostic factors, such as T (AUC values range from 0.502 to 0.808) and N stage (AUC values range from 0.500 to 0.515), the predictive performance of the proposed model was superior in DFS (Fig. S4). Similar results were also detected in PFS and OS prediction (Fig. S4). The improvement of utility when combined circadian clock-based risk score model with T and N stage was limited. Then, for better clinical application, we investigated the power of circadian clock-based risk score model in T2N0 disease and T3-4N1 disease (Fig. S5). AUC values of DFS curves indicated that the risk model performed better in patients with T2N0 stage (vs T3-4N1 stage, AUC value: 0.749–0.834 vs 0.515–0.745, Fig. S5). For PFS, the predictive values of risk score model in T2N0 disease and T3-4N1 disease was almost (Fig. S5). In T2N0 stage, high risk score was significantly related to shorter DFS (p = 0.0024) and PFS (p = 0.034), while no statistical significance was found in OS (p = 0.22, Fig. S6). In T3-4N0 stage, high circadian clock-based risk score was significantly related to shorter PFS (p = 0.016), while no statistical significance was found in DFS (p = 0.12) and OS (p = 0.24, Fig. S6).

We also tentatively applied the risk score model to PFS and OS prediction (Figs. S7–S8). The 3-year and 5-year AUC values of PFS curves ranged from 0.607–0.735 (Fig. S7), while the AUC values of OS curves were higher than 0.700 (Fig. S8). The high circadian clock-based risk score indicated poor PFS (p < 0.0001), and higher 5-year death rate (p = 0.007).

Correlation between clinicopathological parameters and circadian clock-based risk score

The relationship between clinicopathological parameters and circadian clock-based risk scores was evaluated in the TCGA cohort. There was no significant difference in the risk score according to age (p = 0.19; Fig. S9A). However, high risk score was significantly linked to high T status (p = 0.00015) and N status (p = 0.00051; Figs. S9B–S9C). When in comparison with T2N0 stage, we found that higher circadian clock-based risk score was found in the T3-4N1 stage (the TCGA cohort, p = 4e−05; the GEO cohort, p = 3.3e−06; Figs. S9D–S9E).

Functional analysis of circadian clock-based risk score

In the TCGA cohort, on the basis of the circadian clock-based risk score, 246 PC patients were assigned to the high risk score group, while 247 PC patients were assigned to the low risk score group. There were a total of 1114 DEGs between the two groups, including 4643 upregulated and 6471 downregulated DEGs (Fig. 8A). According to the GO analysis results, the top 3 enriched biological processes were ribosome biogenesis (GO:0042254), viral gene expression (GO:0019080), and viral transcription (GO:0019083; Fig. 8B). For molecular function, the top 3 enriched GO terms were small GTPase binding (GO:0031267), Ras GTPase binding (GO:0017016), and structural constituent of ribosome (GO:0003735; Fig. 8B). Moreover, Ribosome (hsa03010, p = 5.15E−15) was also one of the top 10 circadian clock-related pathways in PC (Fig. 8C).

Figure 8 Functional enrichment analysis of circadian clock-based risk score.

(A) The volcano plot visualized a total of 11,114 differentially expressed genes (DEGs) were found between the high and low circadian clock-based risk score groups (p < 0.05). (B) The top 10 circadian clock-related Gene Ontology (GO) terms. (C) The top 10 circadian clock-related pathways were found by the Kyoto Encyclopedia of Genes and Genomes (KEGG) enrichment analyses.

Correlation between CCMCCs and several key prognostic genes

By literature consulting, we noticed some key prognostic genes, such as PTEN, TP53, BRCA1, BRCA2, ATM, RB1, PALB2, CHEK2, MLH1, MSH2, MSH6, and PMS2 (Rebello et al., 2021; Muñoz Fontela et al., 2016; Abida et al., 2019). Then, we explored the correlation between CCMCCs and the mentioned genes (Table S8 and Fig. S10). We firstly evaluated the expression levels of CCMCCs between different mutation status of key prognostic genes in PC (Table S8). Importantly, in the TCGA cohort, the mutation rate of PTEN, TP53, BRCA1, BRCA2, ATM, RB1, PALB2, CHEK2, MLH1, MSH2, MSH6, and PMS2, was 21.26%, 15.99%, 2.23%, 5.06%, 6.07%, 9.72%, 1.62%, 1.62%, 0.81%, 0.61%, 0.81%, and 1.01%, respectively. When compared to the PTEN wild-type group, higher expression of NR1D2 (p = 0.0008006), PRKAA1 (p = 0.00005925), and RORB (p = 0.00000054), were found in the PTEN mutation group (Table S8). When compared to the TP53 wild-type group, higher expression of CRY1 (p = 0.003774), CSNK1E (p = 0.0004053), and RORB (p = 0.006969), were found in the TP53 mutation group (Table S8). In PC, ATM mutation was correlated with high expression levels of CLOCK (p = 0.009059). Mutation status of other genes were also related to CCMCCs’ expression (Table S8).

Moreover, we investigated the correlation between the expression levels of CCMCCs and the above important genes. As shown in Fig. S10, a wild correlation was found among these genes. It is worth mentioning that ATM expression had the highly positive relevance to CLOCK expression (r = 0.633, p < 0.001). A highly positive correlation was also existed between RB1 and FBXL3 expression (r = 0.598, p < 0.001, Fig. S10).

Discussion

Circadian clocks and circadian clock-related genes were essential to maintaining homeostasis. Disruption of the circadian system and aberrant expression of CCMCCs induced tumorigenesis and promoted the proliferation and invasion of cancer cells (Viswanathan, Hankinson & Schernhammer, 2007; Stevens et al., 2014; Papagiannakopoulos et al., 2016; Kettner et al., 2016; Innominato et al., 2009; Huisman et al., 2015). However, CCMCCs expression signature and its function in PC have rarely been investigated. In the research, the expression profiles and functions of CCMCCs were outlined. We also explored the close relationship between CCMCCs and the prognosis of PC patients, thus developing a circadian clock-based risk score model. The circadian clock-based risk score might participate in some biological processes and signaling pathways.

In the present study, we revealed the close relevance between the 22 enrolled CCMCCs and prognosis. Cao et al. (2009) demonstrated that one of the core clock genes, PER1, regulated the expression of androgen receptor, which might affect drug sensitivity in PC. Additionally, in high-grade colon cancer, the relative low expression of PER1 was found (Krugluger et al., 2007). In colon cancer and cholangiocarcinoma, the overexpression of PER1 inhibited tumor progression and growth (Krugluger et al., 2007; Han et al., 2016). In PC, we found a downregulation of PER1 in the T3-4 group in comparison with the T2 group, which was consistent with its expression pattern in other cancer types. In our study, we also detected overexpression of PER1 in the T2N0 stage. Moreover, the expression level of PER1 was positively associated with OS in PC. In stomach adenocarcinoma, patients with high FBXL3 expression showed poor clinical outcome (Liu et al., 2019). However, in kidney renal clear cell carcinoma, an opposite result was found. Specifically, Liu et al. found that patients with high FBXL3 expression showed a better prognosis (Liu et al., 2019). In the 493 PC cases that enrolled in our study, high FBXL3 expression was significantly correlated with longer DFS (p = 0.012) and PFS (p = 0.0066). In breast cancer, overexpression of CSNK1D was found in the N1 group (Abba et al., 2007). Similar overexpression trend of CSNK1D was also found in PC tissues with N1 status. PC patients with short DFS (p = 0.00032), PFS (p = 0.017), and OS (p = 0.016) also showed overexpression of CSNK1D. In glioma, the expression level of BTRC was correlated with clinical outcome (Zhou et al., 2021). Prognostic effects of some CCMCCs in PC remain unclear. In PC, we found higher expression level of BTRC in the T2N0 disease in comparison with the T3-4N1 disease. On the basis of T and N stage, we divided the whole cohort and carried out the survival subgroup analysis. As Tables S2–S7 shown, there were conflicting roles of some CCMCCs in the prognosis of PC patients, such as DBP and NR1D1, suggesting different expression patterns of CCMCCs in T2N0 and T3-4N1 disease. Collectively, in the study, we fully investigated the association between 22 circadian clock-related genes and clinical survival.

As one of the vital biomarkers in PC, high PSA levels also existed in some benign diseases, such as prostatitis, prostatic hyperplasia, and after prostatic massage. Thus, the clinical value of PSA in PC diagnosis and survival prediction remained controversial (Draisma et al., 2003; Manceau et al., 2021; Andriole et al., 2012; Schröder et al., 2012). A multicenter trial reported that PSA failed to affect the prognosis of PC (Andriole et al., 2012), while another experiment found that PSA effectively reduced the death rate of PC patients (Schröder et al., 2012). In our study, the results showed that PSA level at diagnosis was not statistically related to CCMCCs expression. Gleason grade was another important biomarker in PC (Sopyllo, Erickson & Mirtti, 2021; Moris et al., 2020). Recently, a meta-analysis found that Gleason grade was positively associated with recurrence after surgery (John et al., 2021). Gleason scoring system mainly concentrated on the pathological structure of PC, while the transcriptomic and genomic features were missed. In the study, we attempted to find the correlation between Gleason score and gene expression. Nevertheless, a weak to moderate correlation was found between Gleason score and expression levels of CCMCCs. These results might demonstrate that the mechanisms of influences of PSA level, Gleason score, and CCMCCs, on prognosis were different.

Not only clinical factors, but also many genes were considered as vitally prognostic factors in PC, such as PTEN and TP53 (Rebello et al., 2021; Vitkin et al., 2019; Vidotto et al., 2020; Muñoz Fontela et al., 2016). PTEN, one of the tumor suppressor genes, was commonly mutated in PC (Rebello et al., 2021; Vitkin et al., 2019; Jamaspishvili et al., 2018). Several researches found that PTEN mutation or down-expression had extensive influences on tumor microenvironment and PI3K signaling pathways, thus leading to tumor progression and poor prognosis in PC (Vidotto et al., 2020; Jamaspishvili et al., 2018; Garcia et al., 2014; Vidotto et al., 2019; Toso et al., 2014). For instance, PTEN loss promoted T regulatory cell‘ proliferation and infiltration, thereby promoting immune suppression and tumor metastasis (Vidotto et al., 2019). TP53 deficiency was presented in 10% to 40% of PC (Rebello et al., 2021; Muñoz Fontela et al., 2016). TP53 loss aggravated the genomic instability and activated several pathways, thus promoting tumor growth and poor outcome (Bezzi et al., 2018; Robinson et al., 2015; Hamid et al., 2019). In the enrolled cohort, we also found that the mutation rate of PTEN and TP53 was over 10%, which was consistent with previous studies (Rebello et al., 2021; Vitkin et al., 2019; Vidotto et al., 2020; Muñoz Fontela et al., 2016; Robinson et al., 2015; Hamid et al., 2019). An increasing number of studies also highlighted the close relationship between clinical outcome and PTEN, TP53, BRCA1, BRCA2, ATM, RB1, PALB2, CHEK2, MLH1, MSH2, MSH6, and PMS2 (Rebello et al., 2021; Muñoz Fontela et al., 2016; Abida et al., 2019). However, few groups investigated the correlation between CCMCCs and the above prognostic genes. In our research, we considered the mutation status and expression levels of key genes. Through statistical analysis, we found the link between CCMCCs and PTEN mutation, TP53 mutation, ATM mutation, PTEN expression, TP53 expression, ATM expression, etc. These close correlations might also explain why CCMCCs could be used for prognosis prediction.

A certain number of existing prognostic models for PC patients were proposed (Zhang et al., 2020; Xiaoli et al., 2015). These predictive models were developed by lncRNAs, miRNAs, or immune-related genes (Zhang et al., 2020; Xiaoli et al., 2015). Nevertheless, none of them involved circadian clock genes in. The mammalian circadian clock-related genes interacted with each other (Lehmann et al., 2015). CLOCK and ARNTL regulated the activity and expression of NR1D2, one of the nuclear receptors. Subsequently, NR1D2 also could inhibit the mRNA level of ARNTL and NFIL3, leading to the repression of DBP. Collectively, extensive interactions of CCMCCs existed. It is important to make accurate predictions about the prognosis of PC patients for the development of precise treatment. In the present study, through COX regression analysis, we identified 9 vital CCMCCs which could predict prognosis in PC. In unselective PC patients who all received surgery, we developed the circadian clock-based risk score model with higher accuracy in the prediction of DFS than clinical features, including T stage and N stage. Then, we also verified the performance of the risk score model in T2N0 disease and T3-4N1 disease. Importantly, in DFS prediction, the risk score model showed preferable utility in T2N0 disease, indicating that PC patients who were diagnosed at T2N0 stage might be benefited more from the circadian clock-based risk score model. The predictive model also performed well in terms of PFS and OS. The 9-CCMCCs signature also reflected particular molecular functions, cellular components, biological processes, and signaling pathways. Apart from clinical factors, such as T stage, N stage, and PSA, the risk score model might put new insights in the prognosis of PC from the aspect of circadian clock.

We acknowledged some limitations in this research. Firstly, the data of all enrolled public cohorts were obtained retrospectively. Secondly, the heterogeneity of PC samples was nonnegligible. For most PC patients, detailed therapeutic schedules, such as operative approaches and chemotherapy regimens, were unavailable and lacking. Another prospective cohort with less sample heterogeneity and metastasis-free survival data is still necessary. Moreover, in vitro and in vivo experiments are required to further confirm the significant roles of CCMCCs in PC.

Conclusion

In summary, our study improved the understanding of the role of the circadian clock in PC and proposed a circadian clock-based risk score model for prognostic prediction. Moreover, PC patients at T2N0 stage might be benefited more from the circadian clock-based risk score model. These results might be helpful for further investigations of the circadian clock-related molecular mechanisms and the development of therapies for cancer.

Supplemental Information

Supplemental Information 1 Nine core components of the mammalian circadian clock (CCMCCs) differentially expressed between the T2N0 group and the T3-4N1 group (p < 0.05)

Click here for additional data file.

Supplemental Information 2 Correlation among Gleason grade, PSA level, and expression levels of 22 core components of the mammalian circadian clock (CCMCCs) in prostate cancer

Click here for additional data file.

Supplemental Information 3 The survival analysis of circadian clock-based risk score in the GSE70770 dataset

A total of 112 patients with prostate cancer obtained the recurrence free survival (RFS) data.

Click here for additional data file.

Supplemental Information 4 Receiver-operator characteristic (ROC) curves of N stage (A–C) or T stage (D-E) in disease-free survival (DFS), progression-free survival (PFS), and overall survival (OS) prediction

(A) For DFS, 3-year AUC values of N stage in the training cohort and the validation cohort were 0.515 and 0.503, respectively. The 5-year AUC values of the training cohort and the validation cohort were 0.500 and 0.505, respectively. (B) For PFS, 3-year AUC values of N stage in the training cohort and the validation cohort were 0.560 and 0.649, respectively. The 5-year AUC values of the training cohort and the validation cohort were 0.552 and 0.595, respectively. (C) For OS, 3-year AUC values of N stage in the training cohort and the validation cohort were 0.664 and 0.590, respectively. The 5-year AUC values of the training cohort and the validation cohort were 0.613 and 0.541, respectively. (D) For DFS, 3-year AUC values of T stage in the training cohort and the validation cohort were 0.768 and 0.591, respectively. The 5-year AUC values of the training cohort and the validation cohort were 0.808 and 0.502, respectively. (E) For PFS, 3-year AUC values of T stage in the training cohort and the validation cohort were 0.648 and 0.509, respectively. The 5-year AUC values of the training cohort and the validation cohort were 0.602 and 0.600, respectively.

Click here for additional data file.

Supplemental Information 5 Performance of circadian clock-based risk score model in prostate cancer patients at T2N0 and T3-4N1 stage

(A) For disease-free survival (DFS) at T2N0 stage, 3-year AUC values of the training cohort and the validation cohort were 0.749 and 0.766, respectively. The 5-year AUC values of the training cohort and the validation cohort were 0.834 and 0.768, respectively. (B) For progression-free survival (PFS) at T2N0 stage, 3-year AUC values of the training cohort and the validation cohort were 0.560 and 0.597, respectively. The 5-year AUC values of the training cohort and the validation cohort were 0.615 and 0.659, respectively. (C) For DFS at T3-4N1 stage, 3-year AUC values of the training cohort and the validation cohort were 0.745 and 0.515, respectively. The 5-year AUC values of the training cohort and the validation cohort were 0.665 and 0.569, respectively. (D) For PFS at T3-4N1 stage, 3-year AUC values of the training cohort and the validation cohort were 0.500 and 0.508, respectively. The 5-year AUC values of the training cohort and the validation cohort were 0.501 and 0.571, respectively.

Click here for additional data file.

Supplemental Information 6 The survival analysis of circadian clock-based risk score in the the T2N0 cohort (A–C) and the T3-4N1 cohort (D–F)

(A–C) Relationship between circadian clock-based risk score, disease-free survival (DFS; p = 0.0024), progression-free survival (PFS; p = 0.034), and overall survival (OS; p = 0.22) in T2N0 prostate cancer. D-F)Relationship between circadian clock-based risk score, DFS (p = 0.12), PFS (p = 0.016), and OS (p = 0.24) in T3-4N1 prostate cancer.

Click here for additional data file.

Supplemental Information 7 Validation of proposed circadian clock-based risk score model in progression-free survival (PFS) prediction by receiver-operator characteristic (ROC) analyses

(A-B) ROC curves in the training cohort (AUC = 0.607) and the validation cohort (AUC = 0.677) for 3-year. (C-D) ROC curves in the training cohort (AUC = 0.665) and the validation cohort (AUC = 0.735) for 5-year. E) High circadian clock-based risk score was correlated with shorter PFS (p < 0.0001).

Click here for additional data file.

Supplemental Information 8 Validation of proposed circadian clock-based risk score model in overall survival (OS) prediction by receiver-operator characteristic (ROC) analyses

(A-B) ROC curves in the training cohort (AUC = 0.727) and the validation cohort (AUC = 0.805) for 3-year. (C-D) ROC curves in the training cohort (AUC = 0.724) and the validation cohort (AUC = 0.960) for 5-year. (E) The correlation between circadian clock-based risk score and OS (p = 0.13). (F) High circadian clock-based risk score was positively correlated with 5-year death rate (p = 0.007).

Click here for additional data file.

Supplemental Information 9 The correlation between circadian clock-based risk score and clinical features, including age (A, p = 0.19), T stage (B, p = 0.00015) and N stage (C, p = 0.00051). Higher risk score was also found in T3-4N1 stage in the TCGA cohort (D, p = 4e−05) as we

Click here for additional data file.

Supplemental Information 10 Correlation among expression levels of 22 core components of the mammalian circadian clock (CCMCCs), and several key prognostic genes, including PTEN, TP53, BRCA1, BRCA2, ATM, RB1, PALB2, CHEK2, MLH1, MSH2, MSH6, and PMS2,in prostate cancer

Click here for additional data file.

Supplemental Information 11 Characteristics of the TCGA cohort and the GEO cohort

Click here for additional data file.

Supplemental Information 12 Relationship between disease-free survival (DFS) and expression levels of 22 core components of the mammalian circadian clock (CCMCCs) in T2N0 prostate cancer (n = 119)

Click here for additional data file.

Supplemental Information 13 Relationship between disease-free survival (DFS) and expression levels of 22 core components of the mammalian circadian clock (CCMCCs) in T3-4N1 prostate cancer (n = 75)

Click here for additional data file.

Supplemental Information 14 Relationship between progression-free survival (PFS) and expression levels of 22 core components of the mammalian circadian clock (CCMCCs) in T2N0 prostate cancer (n = 139)

Click here for additional data file.

Supplemental Information 15 Relationship between progression-free survival (PFS) and expression levels of 22 core components of the mammalian circadian clock (CCMCCs) in T3-4N1 prostate cancer (n = 28)

Click here for additional data file.

Supplemental Information 16 Relationship between overall survival (OS) and expression levels of 22 core components of the mammalian circadian clock (CCMCCs) in T2N0 prostate cancer (n = 139)

Click here for additional data file.

Supplemental Information 17 Relationship between overall survival (OS) and expression levels of 22 core components of the mammalian circadian clock (CCMCCs) in T3-4N1 prostate cancer (n = 75)

Click here for additional data file.

Supplemental Information 18 Expression levels of CCMCCs between different mutation status of key prognostic genes in prostate cancer

Click here for additional data file.

Additional Information and Declarations

Competing Interests

Author Contributions

Data Availability

The authors declare there are no competing interests.

Wenchang Yue and Xiao Du conceived and designed the experiments, performed the experiments, analyzed the data, prepared figures and/or tables, authored or reviewed drafts of the paper, and approved the final draft.

Xuhong Wang, Niu Gui, Weijie Zhang, Jiale Sun, Jiawei You, Dong He, Xinyu Geng and Yuhua Huang analyzed the data, authored or reviewed drafts of the paper, and approved the final draft.

Jianquan Hou conceived and designed the experiments, analyzed the data, authored or reviewed drafts of the paper, and approved the final draft.

The following information was supplied regarding data availability:

The raw data is available at TCGA (https://www.cbioportal.org/study/summary?id=prad_tcga_pan_can_atlas_2018) and the Gene Expression Omnibus (GEO) Database (GSE70770).

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
