# Peer review of "Prognostic values of the core components of the mammalian circadian clock in prostate cancer"

_PeerJ, doi:10.7717/peerj.12539_

## Round 0.1 · original submission · Major Revisions

Thank you for your submission to PeerJ. Please amend the manuscript based on the comments from reviewers.

·

Basic reporting

The authors evaluated the potential role of the core components of the mammalian circadian clock (CCMCCs) in prognosis prediction in patients with prostate cancer (PC). They found the circadian clock-based risk score model can be useful for prognostic prediction. This is an interesting study for further investigations of the circadian clock‐related molecular mechanisms and the development of therapies for cancer. However, a roubst limitation of this study decreased the importance of the results. The reviewer would like to suggest several key points to improve this manuscript.

Experimental design

#1 The authors need to show the background of patients. It is difficult to find clinical utility from the limited information of the baseline characteristics of patients. The reviewer speculates this study included patients with localized disease alone. But patients with N1 had different treatment strategies and outcomes. The authors need to recognize that the outcome analysis needs a specific population based on the standard of care (such as surgery/radiotherapy for localized disease, systemic androgen deprivation therapy for advanced (N1) disease). The reviewer speculates the clinical implication might be different between the T2N0 disease and T3-4/N1 disease.

#2 The authors need to compare the clinical utility of the circadian clock-based risk score model in combination with robust clinical prognostic factors (Gleason score, T stage, N stage, and PSA, a high-risk localized disease [NCCN or D'Amico criteria]). Again, the key limitation of this study is the mixed population in disease stage and treatment. As it is not demonstrated in the current manuscript, the reviewer could not judge the clinical implication of these scores. The authors need to address this point clearly. Who will have benefit from this circadian clock-based risk score model?

#3 The authors showed many analyses for outcomes. It is confusing. Also, multiplicity issues have arisen in statistical approaches. The authors need to reconstruct the outcomes for the main purpose (ex. 9 of key genes and circadian clock-based risk score model in disease-free survival).

#4 Although the clinical implication of disease-free survival in the localized disease is important, the authors need to show the utility of this model for overall survival or metastasis-free survival, which is the key endpoint of prostate cancer study.

#5 How many patients are included in the survival analysis? The authors need to show the numbers of patients in the figures.

Validity of the findings

It is important that the understanding the role of the circadian clock in PC, but this study could not the rationale of the role of CCMCCs in PC. Many readers would like to know WHY? Although the authors discussed some points in the discussion, the authors had better address the link of other key prognostic genes (such as PTEN, TP53, BRCA, ATM, etc). Otherwise, the clinical implication of the circadian clock-based risk score model for prognostic prediction is weak.

Reviewer 2 ·

Basic reporting

It is an interesting study with novel findings. The authors tried to identify the roles of CCMCCs in prognosis prediction of prostate cancer. Since the molecular pathogenesis of PCa is still poorly known, the identification of novel oncogenes might provide novel possible targets for therapeutics. However, several points of concern, below described, would require further attention from authors and should be addressed before further consideration.

Experimental design

In clinical association part, some other aspects, such as Gleason score, PSA, should be evaluated.

Validity of the findings

Maybe the authors need to confirm the results of some vital CCMCCs in PCa using in vitro or in vivo experiments.

Additional comments

1. The author could introduce some details of PCa in the introduction section.
2. This manuscript focused on the roles of critial clock-related genes in prostate cancer prognosis, the title should be re-written and made clearer.
3. The authors should define the CCMCCs in details in the intoduction section.
4. In discussion section, the authors could provide some discussion on the clinical significance of this study.
5. Several typos have been identified and they should be revised. It might benefit a round of English revision.

---

## Round 0.2 · accepted · Accept

I am glad to accept this manuscript based on the reviewers' comments.

Reviewer 2 ·

Basic reporting

It is an interesting study with novel findings. The authors tried to identify the roles of CCMCCs in prognosis prediction of prostate cancer. Since the molecular pathogenesis of PCa is still poorly known, the identification of novel oncogenes might provide novel possible targets for therapeutics.The authors satisfactorily addressed most of the issues raised by the reviewers.

Experimental design

no comment

Validity of the findings

no comment